# Quantifying online citizen science: Dynamics and demographics of public participation in science

**Bruno J. Strasser**[1]*, **Elise Tancoigne**[2], **Jérôme Baudry**[3], **Steven Piguet**[4],
**Helen Spiers**[5‡], **José Luis-Fernandez Marquez**[6‡], **Jérôme Kasparian**[7], **François Grey**[6],
**David Anderson**[8], **Chris Lintott**[9]

1 University of Geneva, Geneva, Switzerland, 2 Institute of Geography and Sustainability, University of
Lausanne, Lausanne, Switzerland, 3 École Polytechnique Fédérale de Lausanne, Lausanne, Switzerland,
4 Institute of Social Sciences, University of Lausanne, Lausanne, Switzerland, 5 University of Oxford,
Oxford, United Kingdom, 6 Citizen Cyberlab, Information Science Institute, Geneva School of Economics and
Management, University of Geneva, Geneva, Switzerland, 7 Group of Applied Physics and Institute for
Environmental Sciences, University of Geneva, Geneva, Switzerland, 8 Space Sciences Laboratory,
University of California, Berkeley, United States of America, 9 Department of Physics, University of Oxford,
Oxford, United Kingdom

☯ These authors contributed equally to this work.
‡ These authors also contributed equally to this work.
* bruno.strasser@unige.ch

pone.0293289

Educacao Ciencia e Tecnologia Goiano - Campus
Urutai, BRAZIL

**Data Availability Statement:** All data files used in
the figures, as well as all supplementary data are

## Abstract

Citizen scientists around the world are collecting data with their smartphones, performing
scientific calculations on their home computers, and analyzing images on online platforms.
These online citizen science projects are frequently lauded for their potential to revolutionize
the scope and scale of data collection and analysis, improve scientific literacy, and democra-
tize science. Yet, despite the attention online citizen science has attracted, it remains
unclear how widespread public participation is, how it has changed over time, and how it is
geographically distributed. Importantly, the demographic profile of citizen science partici-
pants remains uncertain, and thus to what extent their contributions are helping to democra-
tize science. Here, we present the largest quantitative study of participation in citizen
science based on online accounts of more than 14 million participants over two decades.
We find that the trend of broad rapid growth in online citizen science participation observed
in the early 2000s has since diverged by mode of participation, with consistent growth
observed in nature sensing, but a decline seen in crowdsourcing and distributed computing.
Most citizen science projects, except for nature sensing, are heavily dominated by men, and
the vast majority of participants, male and female, have a background in science. The analy-
sis we present here provides, for the first time, a robust 'baseline' to describe global trends
in online citizen science participation. These results highlight current challenges and the
future potential of citizen science. Beyond presenting our analysis of the collated data, our
work identifies multiple metrics for robust examination of public participation in science and,
more generally, online crowds. It also points to the limits of quantitative studies in capturing
the personal, societal, and historical significance of citizen science.

available from the YARETA database (https://doi.org/10.26037/yareta:lklrzxq3njdhhh4hdzt6nrwdz4).

**Funding:** This study was funded by the Swiss National Science Foundation (www.snf.ch) through the ERC/SNSF Consolidator Grant (BSCGI0_157787), "The Rise of Citizen Science: Rethinking Public Participation in Science," to BJS at the University of Geneva. The funders had no role in study design, data collection and analysis, decision to publish, or preparation of the manuscript.

**Competing interests:** The authors have read the journal's policy and have the following competing interests: CL was co-founder of Galaxy Zoo in 2007 and, as a Professor at the University of Oxford, he works with the non-profit Zooniverse citizen science project (grant). HS is a researcher based at The Francis Crick Institute and collaborates with the non-profit Zooniverse citizen science project (grant). This does not alter our adherence to PLOS ONE policies on sharing data and materials. There are no patents, products in development or marketed products associated with this research to declare.

# Introduction

Since the term "citizen science" was coined in its current sense in 1995 [1,2], scientific research involving the general public has received growing attention from the media, scientific institutions, and science policy bodies [3,4]. The term "citizen science" has now been adopted by numerous scientific organizations in the United States (Citizen Science Association, CSA), Europe (European Citizen Science Association, ECSA), and Australia (Australian Citizen Science Association) and become a commonly used term by science policy and science funding bodies [5]. Citizen science promises to contribute significantly to scientific research, to scientific literacy, and to the democratization of science. It offers an alternative (or complementary) mode of research to the dominant professional sciences. Numerous studies have documented the impact of public participation on science (measured by the number of resulting publications), on scientific literacy (measured by learning assessments), and on democratization (measured by counting the number of participants) [6,7].

Yet, for all the media and scholarly attention to these various forms of public participation in science [7–11], it is still unclear how widespread this phenomenon actually is, how it has changed over time, who are the participants, or what are the most meaningful metrics to make sense of it. A growing number of studies focus on a single online citizen science project, a small set of variables, and a brief time span, however, no study to date has provided a global picture of public participation in science [12]. Without knowledge of the scale and scope of public participation in science it is impossible to assess its historical significance, but also its potential to contribute to scientific research, scientific literacy, or the democratization of science. This paper presents the first global quantitative study of public participation in science which is increasingly subsumed under the label citizen science. Focusing on data from over 14 million participants, it offers a new understanding of the historical dynamics of citizen science and its potential for the democratization of science.

The potential to study public participation in science through big data is still largely underexploited. Yet the digital traces of scientific activities, increasingly carried out online, offer great opportunities for quantitative approaches. Such approaches are rapidly developing for many online activities, for example on social media, but have only marginally been applied to scientific research. Whereas earlier scientometric methods mainly relied on the study of publications and citations [13,14] (and for citizen science see [15]), recent approaches for a "Science of Science" [16] and a "Culturomics" [17] have embraced a larger set of variables. They remain overwhelmingly based on the analysis of texts. Here, along with other authors studying online activities, we look beyond descriptive words to examine individual and collective (online) behaviors. Our reasons for undertaking this data-intensive study are not just to provide a quantitative and qualitative description of the crowds in participatory science. We use this dataset to begin addressing some fundamental questions about the history and the current transformations of science in society and about the potential of citizen science to democratize science and increase scientific literacy.

Public participation in science has a much longer history than "citizen science". It is as old as science itself [18]. Since the Scientific Revolution, different publics with a wide range of occupations have contributed to the production of scientific knowledge, especially by collecting observations and natural objects [19]. But in the nineteenth century, the professionalization of science and the rise of experimentalism created a divide between the professional scientist and the lay public. As an editorial in *Popular Science* in 1902 put it: "the era of the amateur scientist is passing; science must now be advanced by the professional expert." [20]. Since the twentieth century, scientific research has been characterized by the domination of professional scientists and the exclusion of lay people. Therefore, the current claims that

"citizen science", "crowd science", "community science", or "participatory science" are on the rise again deserve scrutiny. A significantly growing involvement of the public in the practice of research could be indicative of a deep historical change in the place of science in society, the forging of a new alliance between professional scientists and lay citizens, or a return to Early Modern configurations of science and society.

Here, we present the first global quantitative study of public participation in science, focusing on citizen science. To examine participant activity we collated a database of citizen science projects since the beginning of its online era (the creation of SETI@home in 1999). Our anonymized database includes data on over 14 million participants who contributed over 2 billion tasks, representing the most comprehensive dataset on participatory science, in terms of data size, temporal scale, and project scope. Indeed, most studies have concentrated either on individual projects or platforms [21–26]; but see [27], at best on distinct modes of participation [28], making our dataset's comparative reach unique. It also differs significantly from most other datasets on public participation which are mostly based on survey data and interviews [27,29–33]; but see [34]. We have followed a different approach because it is often difficult, if not impossible, to control for sample bias in anonymized surveys and to link survey results to online behaviors. Furthermore, "survey fatigue" among participants hinders further data collection [35]. Our database has been constituted through online data scraping, aggregation of public data sets, and data sharing from project managers. However, for ethical reasons, we have strictly limited our data collection to single citizen science projects and have not crosslinked our dataset with data obtained from other platforms, such as social media. Focusing on the most populated citizen science projects, we have collected data from every single participant on each project and, through normalization and aggregation, constituted a dataset that is three orders of magnitude larger than existing datasets based on surveys [30]. The present paper presents a first set of analyses of this dataset.

## Materials and methods

We applied digital humanities methods to collate account data and user profiles from a range of online citizen science projects active from 1999 through 2021 [36]. We subsequently examined multiple aspects of these data, focusing on the participant activity and demographic variables. No datasets contain personal or individual information, only aggregate data about populations.

### Project selection

Based on our typology of citizen science distinguishing five different kinds of participatory practices [18], we selected the three most populous ones for our study, from the creation of the first project in February 1999 up to June 2021: *Computing* (distributed computing project SETI@home and the next nine largest projects on BOINC), *Analyzing* (crowdsourcing projects and platforms Zooniverse, Foldit, Eyewire, and EteRNA) and *Sensing* (distributed observation projects eBird and iNaturalist). In each category, we selected all projects with over 100,000 accounts. These projects cover a wide range of online citizen science activities [18]. In the ten *computing* projects of the BOINC platform, such as SETI@home, participants use their personal computers to perform calculations on scientific data related to a particular field (astronomy, climate science, etc.). Although the main task of these distributed computing projects is performed by a personal computer and not by a human, these online projects produce scientific knowledge with citizens, constitute virtual spaces for science engagement, and offer science learning opportunities, and have thus been considered a major form of citizen science in the media, in policy documents, and the academic literature [4,11,37] see however [38]. In the

*analyzing* projects, participants mostly classify images (nebulae, wildlife, etc.) and transcribe texts (handwritten letters) on the platform Zooniverse or solve three-dimensional puzzles to understand the mechanisms of protein (Foldit) and RNA (EteRNA) folding, or map neurons networks (Eyewire). In *sensing* projects, participants collect observations of plants and animals (iNaturalist) and birds (eBird), or measure environmental parameters (air and water quality, for example). *Self-reporting* (medical self-tracking projects such as PatientsLikeMe; uBiome, etc.), representing the fourth largest population (over half a million accounts in 2018 for PatientsLikeMe alone), was not included due to the sensitive personal data it contained. *Making*, such as do-it-yourself biology (DIYbio), only contributes marginally to the total population of citizen science and takes place mostly offline. It was thus not included in the analysis.

## Data collection

User data typically consisted of a unique username, a date of signup, a date for each task (calculation, observation, classification, etc.), and sometimes a user profile. Online profiles for BOINC projects were initially scraped with the R package rvest in March 2016, for six projects (SETI@home; Rosetta@home; PrimeGrid@home; MalariaControl.net; LHC@home; Climate-Prediction.net), as well as for FoldIt and EteRNA. Subsequent data were collected from the project websites. Datasets for Galaxy Zoo and Zooniverse (Helen Spiers, Grant Miller, and Chris Lintott), Eyewire (Amy Robinson), Massively Multiplayer Online Science (Attila Szantner) were provided by the project organizers. iNaturalist data was assembled by iNaturalist staff from publicly available information and anonymized before delivery to the authors and eBird data was collected from the publicly available eBird platform.

## Data analysis

The data collated was analyzed to identify demographic variables (gender, age, occupation and education, location) and activity variables (signup date, number of tasks accomplished, time at which they were accomplished).

**Gender coding.** For estimating gender, we relied on manual and automatic coding of the unstructured texts of user profiles and usernames (respectively). Manual coding was performed on six random samples of 2,000 profiles each for the top six BOINC projects (S0_Profiles). Between 38% and 55% of the profiles contained gender information, depending on the project (S0_Profiles_Respondants). Unambiguous declarations of binary gender such as "I'm a guy", "a mother", or "a husband" were coded accordingly; ambiguous declarations of partners such as "my wife" were coded probabilistically with an assumption that it designated 99,5% of the time a person of the other sex (according to the US Census data [39]). When first names were given, they were assigned a gender by taking the most probable gender from the names. org database. Non-binary genders were not explicitly mentioned in the profiles and could not be accounted for by automatic coding either (see below). For SETI@home, the result was 92% male (n = 1,106; 95% CI [90.4%, 93.6%]), the five other BOINC projects gave similar results (90–94% male) (S0_Profiles_Gender).

Automatic coding was performed in R by analyzing all usernames (4,754,117, on Feb. 2, 2016) of SETI@home (21% of the usernames contained personal names). We identified potential first names in username strings using a first name databased constructed from census data and assigned a gender probability based on the gender frequency of the name in the database [40,41]. The result was 91.9% male (n = 1,001,913; 95% CI [91.8%, 92.0%]). Manual and automatic coding thus gives almost identical estimates. Crucially, automatic coding can be scaled up and used across platforms to extract gender information from usernames, even when profiles are not existent or inaccessible. To determine gender across all projects

(S0_Profiles_Gender), especially those without profiles, and guarantee comparability, we relied on automatic coding of usernames for all projects (BOINC, Zooniverse, FoldIt, EteRNA, Eyewire, iNaturalist, eBird).

**Age coding.** For estimating age, we relied on semi-automatic and automatic coding of the unstructured texts of user profiles and usernames (respectively).

Semi-automatic coding was performed in R on the same random samples of 2,000 profiles for SETI@home (S0_Profiles). We searched for three broad types of expressions related to age in the user profiles: "X years old/of age/young", "I am X"; "born 19XX"; "born in 19YY", etc. 34% of the profiles contained age-related information detected with this method. Ages below 10 years old and above 100 years old were excluded. They mostly refer to the age of participants' children (e.g. "my 7 y.o. daughter") or buildings (e.g. "living on a 152 year old farm"). Between 19% and 38% of the profiles contained age related information, depending on the project (S0_Profile_Respondants). For SETI@home, we found an average age of 34.7 i.e. 34 years and 8 months (n = 715; 95% CI 0.05 [33,8; 35,7]) (S0_Profiles_Age).

Automatic coding was performed by analyzing all usernames (4,754,117 on Feb. 2, 2016) of SETI@home. Just 0.4% of the profiles contained information likely related to year of birth (e.g. "bob1972"). To obtain the age at the moment of the profile creation, we calculated the difference between the estimated year of birth and the year of the profile creation. The average age was 34 years old (n = 20,559; 95% CI 0.05 [33,8; 34,2]) (S0_Profiles_Age). Because semi-automatic and automatic coding give similar estimates, we applied both methods for projects where profiles were available, and only automatic coding when they were not.

**Occupation and education coding.** For estimating occupation and education, we relied on manual coding of the unstructured texts of user profiles, when available, i.e. "I'm a retired IT sys admin". Only 2% of all users on BOINC have filled out a profile (S0_Profiles_Respondants). We thus checked for potential sample bias by comparing the average credit score, duration in the project, and team participation (yes or no) of the users with and without profile. Participants who complete a profile are overall more invested in the project [42], but otherwise have similar characteristics. Unlike surveys, coding of online profiles makes it possible to estimate sample bias and create much larger samples.

As automatic approaches failed to give us an estimate of participants' profession and education [42], we coded manually the same random samples of 2,000 profiles for each of the six BOINC projects as well as for Foldit. We coded all expressions referring to profession and education into three categories: "science and engineering", "IT" and "other". Between 30% and 36% of the profiles contained information on professions, and between 5% and 12% on education, depending on the project (S0_Profiles_Respondants).

**Lorenz curves.** To visualize the distribution of work among participants, we plotted Lorenz curves and calculated Gini indexes. Lorenz curves were calculated by taking the cumulative contributions on SETI@home; Galaxy Zoo, and iNaturalist for 30 days in November 2019. Gini indexes were calculated as the ratio of the area between the Lorenz curve and the diagonal (equal distribution) and the total area under the diagonal [43]. We plot the cumulative share of the active population vs their cumulative contribution in the relevant activity metric (credits, classifications, observation, respectively).

**Active users.** The number of active users was estimated by taking all unique users who contributed at least one task in a 30-day period ending in June 2021. The value for BOINC was taken from boincstats.com; for Foldit from fold.it; for Eyewire from eyewire.org; for EteRNA from eternagame.org; and for iNaturalist from inaturalist.org/stats, all accessed on June 25, 2021. Data for Zooniverse was provided by the project organizers. For eBird, we identified the number of unique user id providing at least one observation during the period from the entire observation dataset available online (**http://science.ebird.org**). For iNaturalist, observations,

but not classifications or comments, were considered a task, since the project was taken as a *sensing*, and not an *analyzing* project.

**Mobility among sub-projects on the BOINC platform.** The mobility among projects was calculated by taking the first project on which the user signed in and the second to which he/she subsequently contributed. The visualization was created using the Circos software [44].

**Countries.** The administrators of BOINC, Zooniverse, and iNaturalist provided the location of all active participants from September 1st to September 30th 2019 by geocoding the IP number of each active participant and attributing a country with GeoPy (geopy.readthedocs. io).

**Time.** The time at which an online activity took place was determined by taking the time stamp at the creation of the user account on BOINC. The time zone was determined by reference to the country given for the location. For the United States, the time zone was estimated by assuming the location followed the population distribution in the country, i.e. 47.6% for Eastern Time Zone, 29.1% for the Central Time Zone, 6.7% for the Mountain Time Zone, and 16.6% for the Pacific Time Zone (i.e. an average time zone of GMT—5.95). We excluded data from Russia due to the difficulty of assigning one of the eleven time zones in the country.

S0_Datasets provides a summary of all datasets used in this paper.

## Results

### The size of the crowd

Since 1999, when the first online participatory project was launched (SETI@home), over 14 million people, as estimated by the number of user accounts, have contributed to citizen science as *computers*, *analyzers*, or *sensors* within the 16 online projects analyzed here, representing the largest citizen science projects worldwide (Fig 1). By far, the most common mode of participation has been distributed computing, i.e. sharing personal computing power to perform scientific calculations and the most popular project of this kind was SETI@home. Just a month after being launched, in April 1999, 350,167 new users had joined SETI@home. Within six months, over one million individuals had created an account on SETI@home. Although over six million individuals would eventually join the project over its lifespan (April 1999—March 2020), the initial growth rate continuously decreased. Such a massive influx of participants in a citizen science project was unprecedented—and never repeated since by any other participatory project.

This result, alone, already challenges the common narrative about the recent "rise of citizen science" [45–47] or "rise of the citizen scientist" as an editorial in *Nature* put it [48]. Online citizen science was growing faster two decades ago than it does today (Fig 1) and, for the most populous citizen science project, SETI@home, the number of active participants peaked in 2000 and has declined ever since (Fig 3). But this peak in public participation is largely due to a single project with an unmatched number of users: SETI@home. For this reason, we analyzed further the different modes of public participation in science. They exhibit widely different historical dynamics (Fig 1). *Computing* grew exceptionally rapidly starting in the late 1990s, carried by the success of SETI@home but has leveled off around 6.4 million participants. *Analyzing* (or crowdsourcing) rose a decade later, at the end of the 2000s, carried by the launch of Galaxy Zoo in 2007 (participants classify images of galaxies), of Foldit the same year (participants fold proteins in 3D), and of the crowdsourcing platform the Zooniverse in 2009 (hosting a variety of projects like Galaxy Zoo). More recently, since 2016, the Massively Multiplayer Online Science (MMOS) crowdsourcing projects embedded in the EVE-online game, a popular online role-playing game with a community of over half a million gamers, briefly brought several hundred thousand participants to analyze scientific data [49]. Yet, the rate of growth of

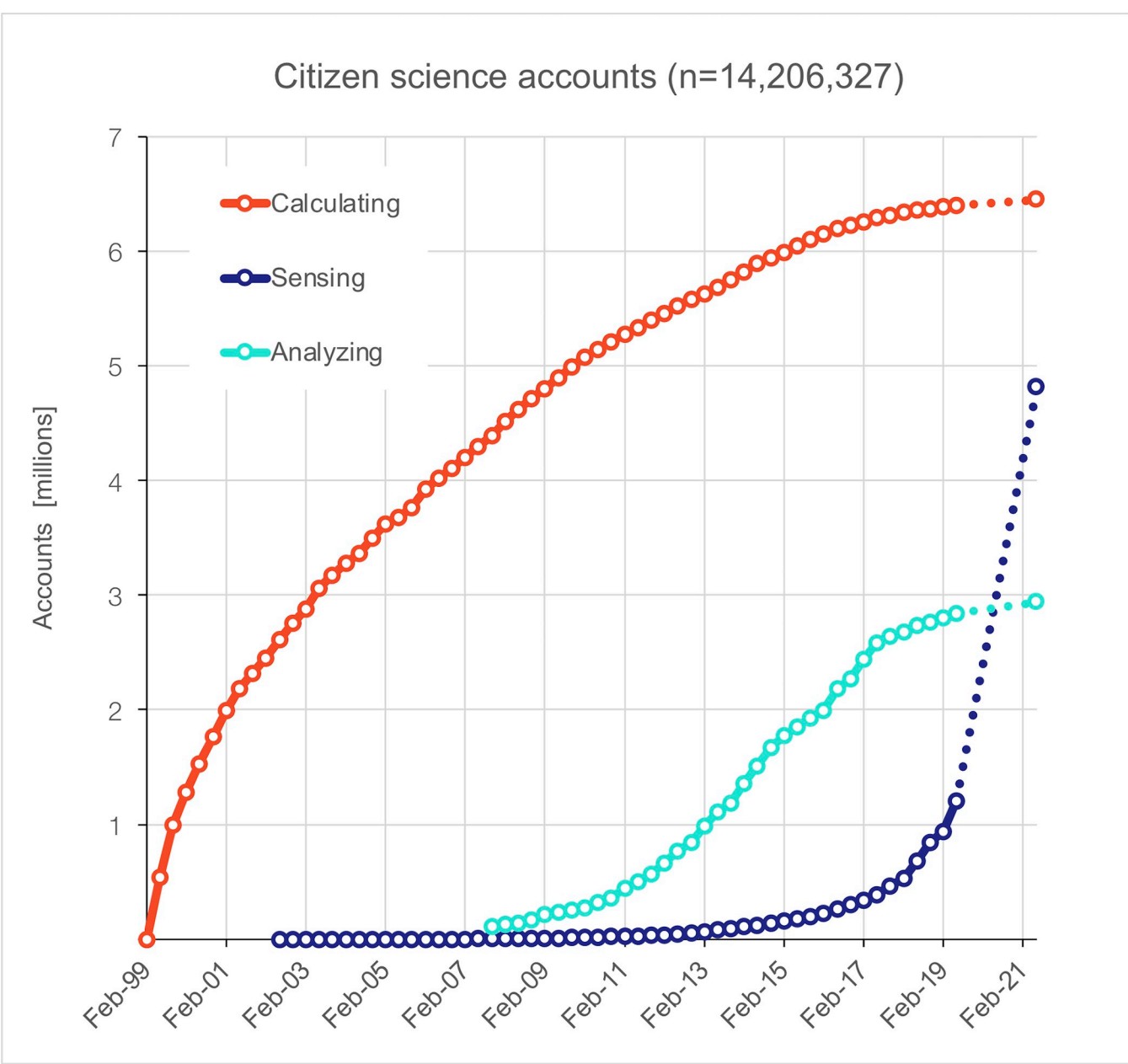

**Fig 1. The number of individual accounts created since 1999 (the beginning of SETI@home) on online citizen science projects in three different categories (*Computing, Sensing, Analyzing,* see text).** The three categories show strikingly different dynamics, with almost no growth for *Computing*, but increasing growth for *Sensing*. The dashed lines represent a linear extrapolation between the last two data points in the dataset (the gap is due to the unavailability of data in that time period).

*Analyzing* has continued to decrease, with the total number of users signed up leveling off around 3 million. By contrast, *Sensing* (or collecting observational data) has followed an exponential growth (approx. 18 months doubling time) carried by nature observation platforms such as iNaturalist (created in 2008) or eBird (created in 2002). These projects reached 1.4 million users in 2019, surpassed 4.8 million by June 2021, and were continuing to grow rapidly. In the history of online citizen science, these sensing projects are fueling a *second* growth period, after the initial rise due to SETI@home around year 2000.

**Table 1. Number of individual accounts created on citizen science platforms (from platform launch to 2021) and the number of active accounts in June 2021.**

| | | Creation | Total accounts | Active accounts | % active |
|---|---|---|---|---|---|
| *Computing* | | | **6'396'956** | **71'188** | 1.1% |
| | **BOINC (w SETI@home)** | 1999 | 6'396'956 | 71'188 | 1.1% |
| | | | | | |
| *Analyzing* | | | **2'964'079** | **22'541** | 0.8% |
| | **Zooniverse** | 2007 | 2'326'955 | 20'748 | 0.9% |
| | **EyeWire** | 2012 | 266'635 | 714 | 0.3% |
| | **EteRNA** | 2009 | 251'306 | 614 | 0.2% |
| | **FoldIt** | 2007 | 119'183 | 465 | 0.4% |
| | | | | | |
| *Sensing* | | | **4'814'466** | **665'466** | 13.8% |
| | **iNaturalist** | 2007 | 4'118'955 | 323'476 | 7.9% |
| | **eBird** | 2002 | 695'511 | 341'990 | 49.2% |
| | | | | | |
| **TOTAL** | | | **14'175'501** | **759'195** | **5.4%** |

Although the number of accounts is the most used metric to measure the size and the success of online projects and platforms across the internet, it is misleading as a measure of actual participation. Indeed, only a fraction of accounts actually represents active participants [29]. For instance, out of the 533,322 accounts created on Foldit, just 119,183 (22%) had completed at least one task (puzzle solving). The 14 million users accounts created on citizen science platforms thus represents a gross overestimate of the actual population of participants.

A far more meaningful metric than the number of accounts, inflated by spam and inactive users, is the number of participants contributing to a project in a given time frame. Of the 6.4 million users who signed up and completed at least one task on the *computing* projects such as SETI@home, 71,188 (1.1%) were active in June 2021, twenty years after the launch of the project (Table 1). For more recent projects, such as the *analyzing* platform Zooniverse, out of 2.3 million once active users, around 20,748 (0.9%) were active in June 2021, and on Foldit, just 465 (0.4%) users were still active. *Sensing* projects have a larger fraction of active users. Of the 4.1 million once active users on iNaturalist, 323,476 were active (observers) in June 2021 (7.9%) and for eBird, 341,990 were active (49%) in June 2021 out of 0.7 million once active users (Table 1). The fraction of active participants in citizen science projects thus varies considerably, with the monthly active population representing anywhere from a fraction of one percent to close to 50% of the total number of participants who were once active. This demonstrates that the total number of accounts is a poor indicator of participant activity.

In June 2021, there were 23k persons *analyzing*, 71k *computing*, and 665k *sensing*, or three-quarter of a million individuals overall contributing to citizen science projects. To our knowledge, this is the first quantitative estimate of the global monthly participation in citizen science. This number slightly overestimates the actual number as some users might be counted multiple times if they contribute to several projects on different platforms. But it has excluded all accounts with zero contribution (mostly spam). Although this study is not exhaustive, it includes all the largest citizen science projects and platforms, thus capturing the vast majority of online participants (see discussion). Participation to nature observation projects (eBird and iNaturalist) shows seasonal variations, with June being one of the most active months. Hence, it is reasonable to say that there are under one million monthly participants in citizen science year-round.

In addition to distinguishing active participants from inactive accounts, one can further characterize participation by measuring the distribution of work among active participants for a given project. Within online communities, participation has often been described as particularly uneven, with a small number of individuals performing the largest part of the collective work, sometimes estimated as a power law, or colloquially as the "90-9-1" rule, with 90% viewing, 9% editing, and 1% creating content [50]. Studies of the crowdsourcing projects available on Zooniverse have confirmed this picture, using standard economic metrics to measure income inequality, the Lorenz curves and Gini coefficient [34,51]. *Analyzing* (measured through the Zooniverse project) has the highest inequality, with the top 10% contributing 79% of the total tasks (and the lowest 50% contributing 2.7%, Gini = 0,84) (Fig 2). *Computing* (measured through the SETI@home distributed computing project) is slightly more egalitarian, with the top 10% contributing 71% of the total work (and the lowest 50% contributing only 2%, Gini = 0,82). In this case, the work is performed by the participant's computers, with some individuals having a single laptop, while others having an array of overclocked computers [6,37]. *Sensing* (the iNaturalist platform) has a similar division of labor for its observations, with the top 10% contributing 69% of the work (and the lowest 50% contributing 6%, Gini = 0,76), illustrating not only different engagement intensities, but also qualitatively different modes of engagement [52]. These results are comparable to previous estimates of work distribution on citizen science platforms [51,53], but studies of *analyzing* projects on the Zooniverse platform show there are also large differences between individual projects [34].

Finally, the population contributing to a given project can be characterized by the distribution of seniority among participants (Fig 3A and 3B). The collective level of skill that participants bring to a given project depends not only on their prior education-based expertise, but also on the experience they gain through repeated practice in the project. Participatory projects who succeed in retaining participants will see the total number of years of experience in their population increase over time, with potentially beneficial consequences on the number of tasks performed and on data quality. In the case of SETI@home, the total number of years of experience of the active population remained approximately constant until around 2013, even though the absolute number of active participants decreased. In the case of SETI@home, "experience" mostly involves the technical skills mobilized to overclock computers and maximize their data crunching power. In other projects, such as analyzing images on Galaxyzoo, "experience" involves the cognitive skills used to identify patterns and identify unusual images.

Online participatory projects have a high turnover, like other online communities such as Wikipedia [54,55], i.e. less than half of the participants who signed-up on citizen science platforms and contributed at least a single task are still active after one year. But participants who stop contributing to a citizen science project sometimes join another one, a shift made easier by the platformization of citizen science (Fig 4). For *computing*, 36% of the once active participants on SETI@home joined another project on the BOINC platform and more than 66% of the participants to Milkyway@home had first joined another project (mostly SETI@home).

## The demographics of the crowd

The OECD scientific workforce in 2020 was 40% female and had a median age of 43 years old, whereas census data for the OECD population was 51% female and 40 years old (OECD 2021). Is the population participating in online citizen science more representative of the general population than the (professional) scientific workforce, thus contributing to its "democratization"?

The cross-platform results of public participation in science demographics are based on the study of over 14 million participants. However, for some variables, such as gender and age,

## Online work distribution in citizen science projects

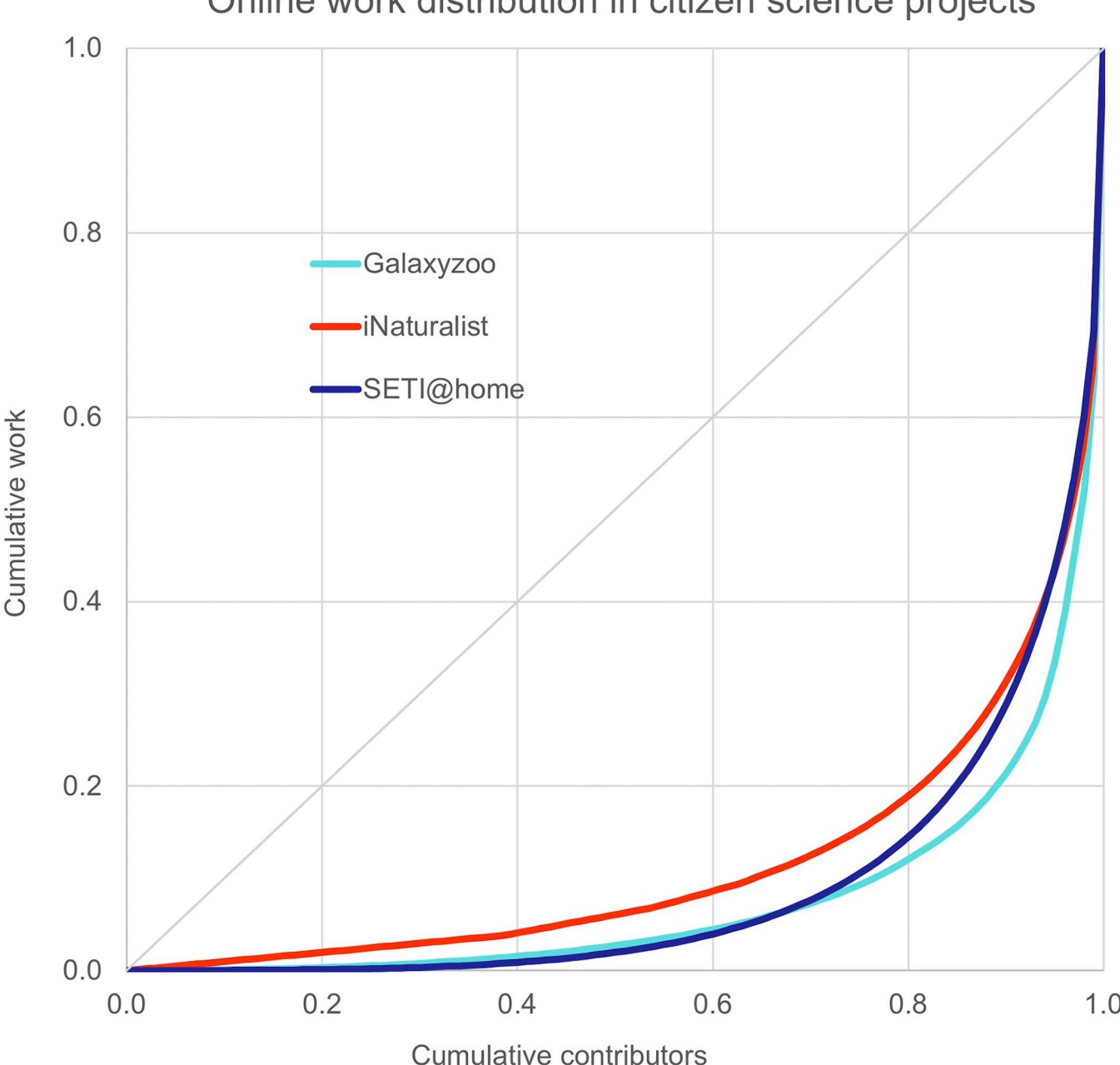

**Fig 2. Lorenz curve showing the distribution of work (cumulative share of contributors vs cumulative share of work) within three types of citizen science projects:** *Computing* **(SETI@home),** *Analyzing* **(Galaxyzoo) and** *Sensing* **(iNaturalist).** The diagonal represents an equal contribution by all participants. All projects exhibit a similarly high inequality of work distribution.

data was only available for a smaller population, 1,321,233 and 26,840 participants respectively. Our analysis confirms unambiguously that participants, for a wide variety of projects, from *computing* to *analyzing*, are male dominated (52–92% male), but that there are large differences among types of citizen science projects (Fig 5). This gender imbalance is most pronounced for *computing* projects, such as SETI@home, which is over 90% male, but less so for *analyzing* projects such as Galaxy Zoo which is 68% male. On the other hand, *sensing* projects,

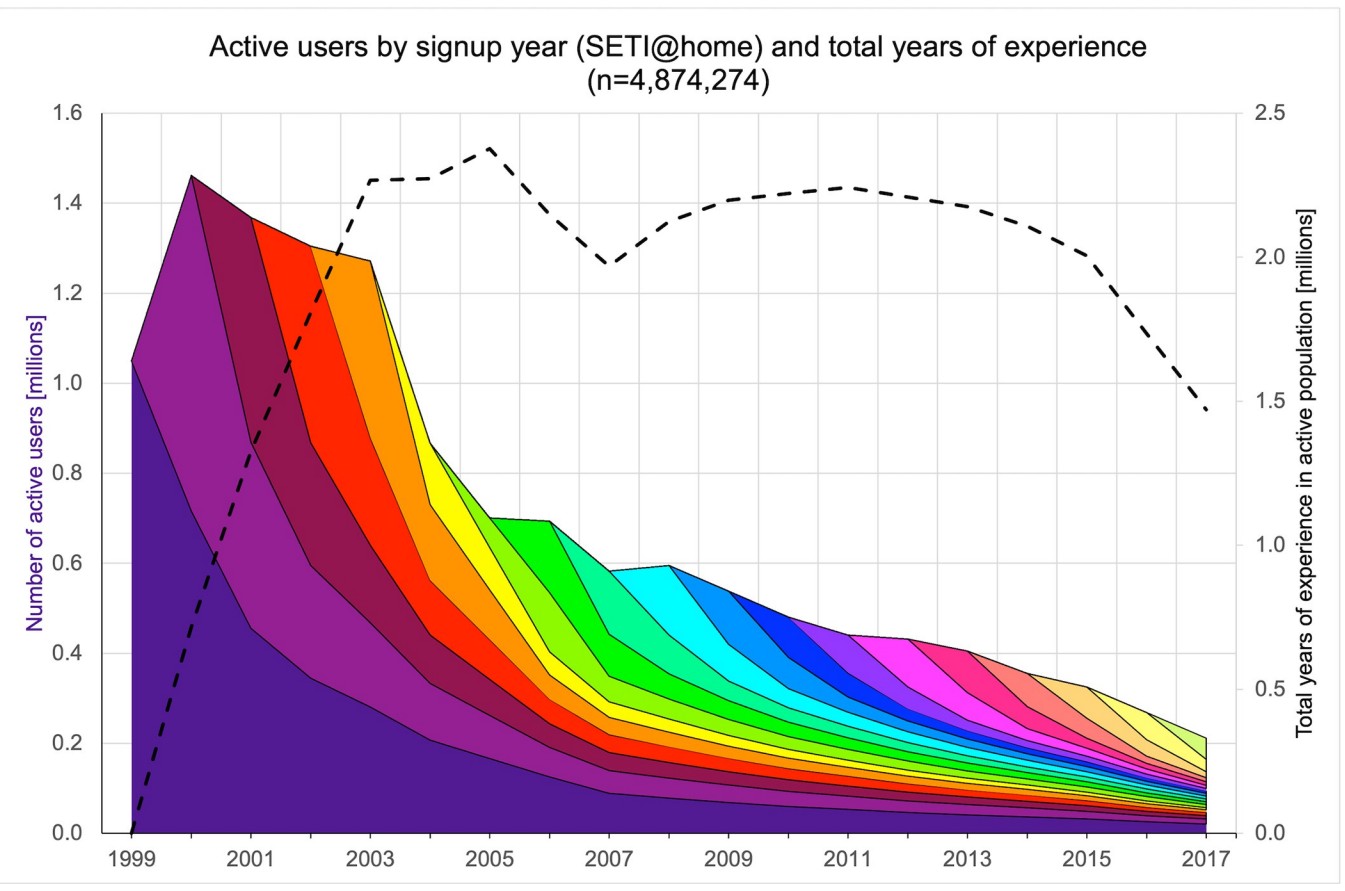

**Fig 3. Number of monthly active users on SETI@home over time, stratified by their signup year (seniority, or years of experience).** The lower strata represent users with more years of experience than the higher ones. The dashed line represents the total number of years of experience of the entire active population. The collective experience of the population peaks at a later time than the number of participants.

such as iNaturalist, are almost on parity, with 56% males (51% for eBird). Except for *sensing* projects, the current online citizen science population is thus *less* representative, in terms of gender, of the general population, than the current population of professional scientists. However, with the rise of *sensing* projects, the population of online citizen scientists is moving towards greater gender equality.

The age distribution and the median age of citizen science participants is also markedly different from either the general population or the professional scientist's population, and there are large differences among projects (Fig 5). For the *computing* project SETI@home, only 9% of the participants are between 10 and 19 years old (median age = 34), but 26% for the *sensing* project iNaturalist (median age = 22). Expectedly, citizen science projects that include opportunities for school students, such as iNaturalist, have a higher representation in that age category (40%). At the same time, 12% of the participants to iNaturalist are 60 years old and over, showing that such projects offer opportunities for a very broad age range. *Analyzing* projects, such as Galaxy Zoo, also draw many school-age participants (35%), usually unaccounted for in surveys, but far fewer participants over 60 years old (4%). Heavily gamified *analyzing* projects, such as FoldIt, have an even narrower age base: 65% of the participants are under 30 years old (median age = 25) and only 3% are over 60 years old. The median age of all projects is lower than the median age of the general OECD population (40 years), or the Western world

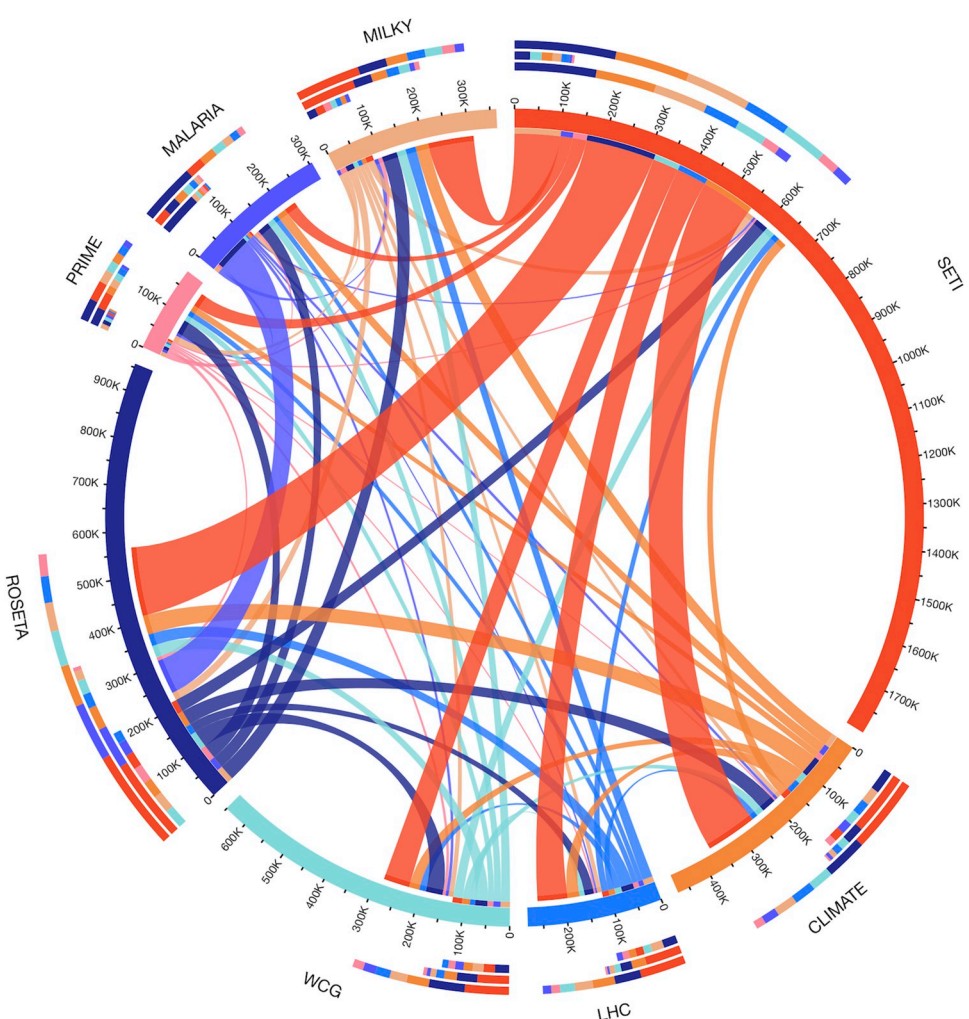

**Fig 4. Participant flow from one distributed computing project to another on the BOINC platform.** The outer circle represents the first project that participants joined. Some projects are major participant "donors" (SETI) and others are "receivers" (PRIME). The participant flows do not follow disciplinary or thematic boundaries (physical sciences, life sciences. for example).

scientific workforce (43 years). However, their broader distribution is more representative of the general population since it includes 11% of 10–20 years old (34% in the general population) and 17.63% over 65 years old (17,20% in the general population), which are obviously almost absent in the professional workforce.

The third, and arguably the most significant, demographic component of participation is the educational and professional background of the participants. The assumption behind most news stories about citizen science is that participants are "ordinary citizens" or "members of the general public" as the Oxford English Dictionary puts it in its definition of citizen science (OED 2021). But our analysis of profiles of participants across the six largest *computing* projects paints a different picture: among those who mention their occupation, over 60% are in the field of science or IT, and an additional 20%, who are professionally employed in a different sector, have a college level background in science or IT (participants with science occupations might be more likely to report it, but not necessarily, as the organizers of citizen science project have valued the participation of "ordinary citizens".) For *analyzing* projects, biographical

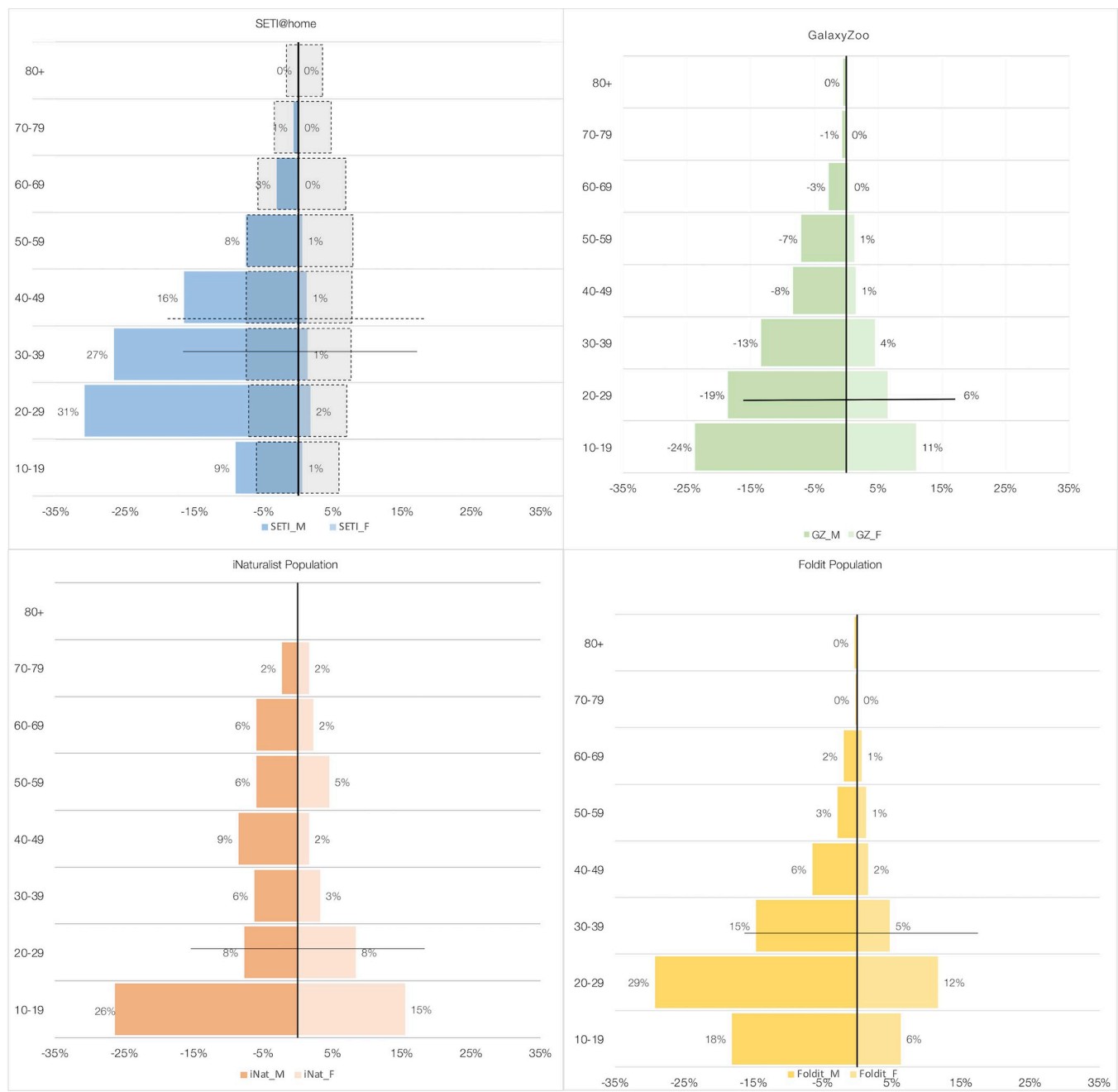

**Fig 5. Demographic pyramids for selected citizen science projects (female on the right side).** In grey, on the first graph, the age and gender pyramid for the OECD population in 2021. The horizontal line represents the median age. The pyramids show a strong over-representation of men (less pronounced for iNaturalist) and a far younger participant population than the general population.

profiles available on FoldIt show that 80% have an occupation—and 90% a background—in science or IT, confirming the trend observed for *computing* projects. The *analyzing* projects on Zooniverse have no biographical profiles, but earlier surveys indicate that the level of education is significantly higher than the general population, since over 70% of survey respondents have a bachelor's degree [25,25,56,57]. For *sensing* projects, such as iNaturalist or eBird, self-descriptions center on the affective dimension of amateur naturalist pursuit, and rarely

mention occupation or education, preventing a comparison with the other projects. However, several surveys of participation in environmental citizen science show that a similar trend is at play there [26,28,58–60]. This high level of scientific literacy and expertise among participants may contribute to the quality of data collection and analysis but indicates that online citizen science is making a more limited contribution to democratizing science than usually claimed.

## The spatial and temporal dynamics of the crowd

Citizen science is often portrayed as a global phenomenon. Indeed, participants to the *computing* project SETI@home declare over 200 countries of residence and participants to the *analyzing* projects on the Zooniverse platform can be localized in 196 countries. Yet a closer look at the spatial distribution of participation offers a more complex picture. In 1999, 59% of active participants on SETI@home declared residence in the United States and 32% in Europe, i.e. over 90% are in OECD countries. Five years later, and to the present day, around 40% of participants declared residence in the United States, 40% in Europe, and 6% in Asia (with the remaining locations being unknown). *Computing* witnessed a rapid Europeanization, along with the adoption of faster broadband connections, but has not broadened its geographic scope since 2004, remaining predominantly located in the United States and Europe. The prevalence of the United States (59%) and European countries (25%) is also clear for the projects, such as Galaxy Zoo, on the Zooniverse platform, and for the *sensing* project iNaturalist (51% and 20%) (S6_Country).

At a finer scale, there are large national differences in terms of participation per inhabitant (Fig 6), which reveal that not only geography, but also language, is a major factor for participation (even when translated interfaces become available). For the *analyzing* projects on Zooniverse, six of the top-ten countries are English-speaking: the United States (1), UK (3), New Zealand (4), Canada (5), Australia (6) and Ireland (9). Some individual countries have a markedly higher densities of participants per inhabitant than their neighbors, such as the Netherlands (2), which is unsurprising given that it has the highest English proficiency in Europe [61]. More unexpectedly, Uruguay has the seventh largest density of participation per inhabitant for *analyzing* and the absolute highest for distributed computing, with over two participants per one thousand inhabitants. Uruguay has not only the highest literacy rate in Latin America, but has also been a testing ground for the MIT's One Laptop per Child (OLPC) project since 2007 [62]. As for the *sensing* project iNaturalist, countries with particularly rich biodiversity, such as Costa Rica (6), Panama (7), and Taiwan (9), stand out with high densities of participants per inhabitant (participants may be visitors rather than residents).

In addition to the spatial structure of participation, one can gain some insights into its temporal structure. Public participation in science is often considered to belong to the category of "serious leisure" [63], i.e. an unpaid recreational activity which is pursued systematically, possesses a significant learning dimension, and is carried outside of regular working time. As such, it contrasts with professional activities which not only generate income but are mostly carried out during weekly day-time hours, from 9 am to 5 pm, and typically later for scientific research. The temporal distribution of public participation in science presents a different picture (Fig 7). Signup times (when people created their user account) for the *computing* project SETI@home take place 24/7, not only on evenings and weekends. It peaks however between 6 pm and 9 pm on weekdays, Mondays through Thursdays, but also on Sundays. Friday and Saturday evenings have a slightly lower level of participation. There are no major differences in the temporal distributions of participants in the United States, Europe, and Asia.

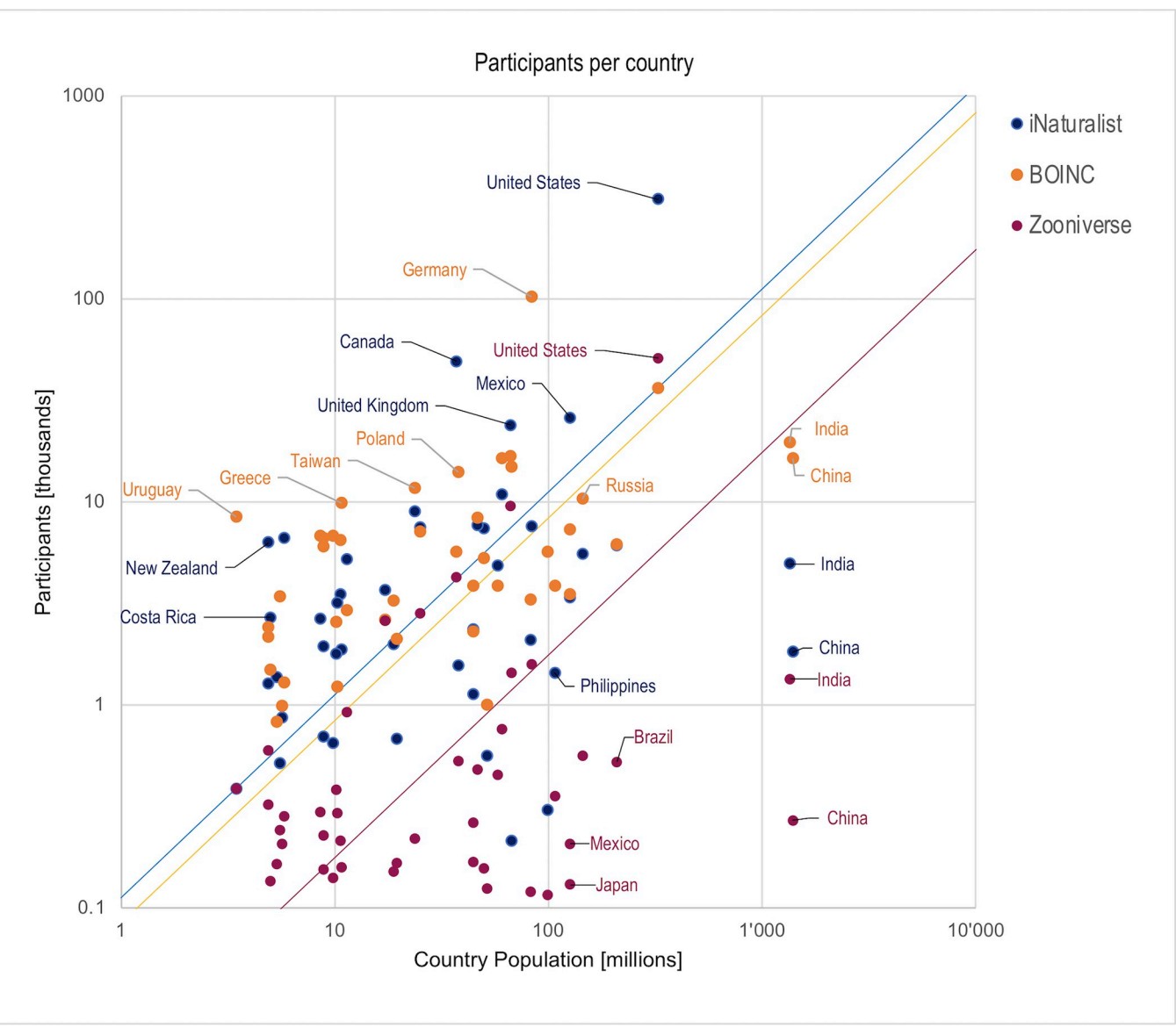

**Fig 6. Log-log plot of the total number of participants in citizen science projects vs country population in selected countries.** Distance from the data point to the average line (perpendicularly) indicates the difference above (or below) average participants per inhabitant in all countries. The largest contributors to citizen science projects (> 10,000 participants per project) are all European or North American countries. Some countries, such as New Zealand and Costa Rica for iNaturalist and Uruguay for BOINC, have an exceptionally large number of participants who declare their residence in these countries (per inhabitant). See supplementary data S6_Country for full table.

## Discussion

Here, we present the analysis of a unique collection of online citizen science project data, collated and shared for the first time. This data includes a wide variety of projects, from bird watching to distributed computing, and is representative of the contributions of over 14 million participants over a period more than two decades. Prior work in this arena has typically focused on participant motivation, learning, and data quality; often the most useful metrics for the managers of citizen science projects [6,11]. Here, we instead examine the data collated in the context of broader historical and sociological questions, to gain quantitative understanding

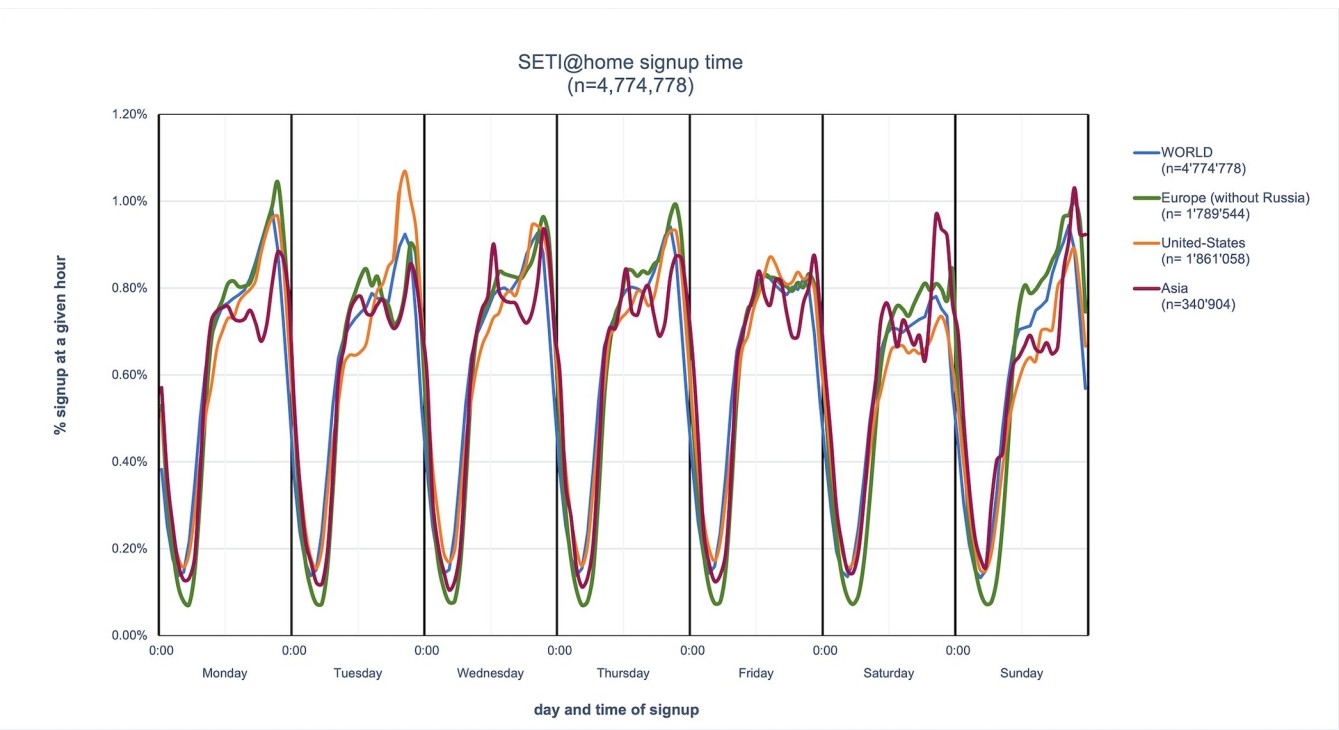

**Fig 7. Day and local time at which participants created accounts on SETI@home.** In all regions, people signed up every day of the week, at all times, with a peak around 9 PM. There are no marked regional differences. Friday and Saturday evenings are less active than Sunday through Thursday evenings.

of the current "participatory turn" [64] in scientific research. Our analysis addresses three major questions; first, how large is the population of participants in citizen science, as measured by active participants online? Second, is citizen science, measured in terms of active participants online, expanding? Third, is online citizen science making science more democratic, i.e. demographically more similar to the general population? Finally, we highlight the limitations of quantitative studies in capturing the personal, societal, and historical significance of citizen science.

Our first conclusion is that approximately three-quarters of a million people, world-wide, contributed to an online citizen science project in June 2021. The actual number of participants is obviously higher, as our sample only includes the largest online participatory projects. However, we believe it provides a good approximation of the total number of online participants. SciStarter, the most comprehensive citizen science platform, hosted just over 1'500 projects worldwide, but most of them are local in scope and have small numbers of participants signed-up (less than one hundred), and only a fraction of them is active in a given month (https://scistarter.org, accessed June 7, 2021). Some very large projects, such as the British Nature Observation Project OPAL, have not been included in our survey, because their participant metric rests on a different definition from ours, i.e. anyone who has had a learning experience in their outreach activity, not the actual number of participants contributing scientific data [65]. The big picture is that the estimates of "millions of participants" engaged in "collecting and/or processing data" [66] or "millions of people" engaged in "a single project" [67] constitute an overestimate. Globally and across all online projects, in any given month in 2021, it is a safer to say that there were less than one million people contributing to citizen science projects, across *Computing*, *Analyzing*, and *Sensing*. But what is the significance of this number?

Since one of the three goals of citizen science is to contribute to scientific research, one should compare the number of citizen scientists to the professional scientific workforce. According to UNESCO data [68], there are around eight million professional researchers worldwide full time equivalents in 2013, thus citizen scientists represent around 10% of that number (these categories can overlap), not an insignificant number. Of course, this proportion will vary widely between fields, being higher in ornithology than molecular biology, for example. Furthermore, since professional researchers often work full time, but citizen scientists typically only spend a few hours a week on research, the hourly contribution of citizen scientists to scientific research is more likely to be less than 1% of the professional science workforce.

But such comparisons have their limitations. For most of the projects carried out through public participation, there is little evidence that they would be carried out by professional scientists alone. It thus seems doubtful that citizen scientists are replacing the professional scientific workforce [69]. On the contrary, the availability of citizen scientists makes certain kinds of studies, such as large-scale ecological surveys, possible [70,71]. Without citizen scientists, studies of common birds present in backyards could hardly be carried out. Although the public seems indispensable for a number of *sensing* projects, for *analyzing* projects, especially of the crowdsourcing kind, the conversation is now about how human participants can be replaced by AI and other automated data analysis methods [11,72]. The growing integration of AI in citizen science is already transforming the field. Since citizen science data is now used for AI training, the results presented here draw attention to the potential biases introduced by the uneven distribution of participant demographics and the necessity to monitor the profiles of participants. In the near future, AI could also reduce significantly the need for citizen scientists [73]. Since AI can perform some of the tasks more efficiently than humans, it could impact the historical trends described in this paper, especially for tasks such as analysis. As for *computing*, in 2022 the entire calculating power of the BOINC distributed computing platform, 15 Peta-FLOPS over 24 hours (https://boinc.berkeley.edu/, accessed Nov 28, 2022), was still on par with the fastest single supercomputers and thus offers a significant opportunity to perform scientific calculation at a marginal cost (which is distributed over all volunteers).

Another stated goal of citizen science is to contribute to science education [31,66]. Although our study does not address directly learning outcomes, one can contextualize the total number of participants to online citizen projects with other forms of public engagement and outreach with science. Major natural history museums and science centers typically attract several million visitors every year: in 2014, 7.3 million at the National Museum of Natural History in Washington DC and 3.2 million at the Cité des sciences et de l'Industrie in Paris [74,75]. Worldwide, science centers, natural history museums, botanical gardens and other institutions devoted—at least in part—to science outreach welcome over 100 million visitors annually [18], at least one order of magnitude above the number of individuals engaged in online citizen science. However, citizen science projects offer opportunities for long term engagement and a qualitatively different (possibly more active) learning experience [76].

In terms of public engagement with science, the global readership of popular science magazines offers another point of comparison. At least two magazines have a readership above one million: *National Geographic* (6 million globally) and *Popular Science* (1.4 million). A few science magazines have a circulation in the hundreds of thousands, in the United States (*Scientific American*; *New Scientist*, *Discover*), France (*Science & Vie*, *Science & Avenir*), Italy (*Focus*), and Brazil (*Superinteressante*). The online audience of printed media, through science websites and social media is typically one or two orders of magnitude higher, with *National Geographic* claiming an online audience of 433 million households [77], although the number of active followers is lower. Around one dozen popular science websites have more than one million monthly visitors [78], with Howstuffworks at close to 20 millions and NASA at 12 millions

visitors. Several dozen science videos on Youtube have been viewed over 10 million times, with some peaking at over 100 million views. The population of individuals who are engaged with science content online every month could well be above 100 million. Arguably, many of these individuals experience different learning opportunities than participants in citizen science projects, where they may acquire not only scientific content knowledge, but also knowledge about the nature of scientific inquiry [79].

In short, for every active online citizen scientist, there are ten visitors to science outreach institutions, and one hundred visitors of online scientific content. Yet, although citizen science currently only represents a minor form of public engagement with science, it offers a qualitatively different kind of experience and educational opportunity. Its value and significance can thus not be fully captured by the number of its practitioners alone.

Our second major question pertains to the dynamics of citizen science: is active participation expanding? Our results support a more complex narrative than a simple "rise of the citizen sciences": the number of new participants per month is declining for the *computing* and *analyzing* projects, but is increasing for *sensing* projects. The number of active participants has also declined between September 2017 and 2019 for the first two categories but rose sharply for the last. The stellar rise of participation on SETI@home in 1990 (one million participants in six months) has never been repeated for thirty years (SETI@home closed down in 2020). In 2020, another citizen science project succeeded in enrolling over one million participants in just three months by embedding a science puzzle into Borderlands 3, "a fast action role-playing first-person video game", with an *existing* userbase of several million players [80]. But participants joined Borderland 3 to play an exciting game, not to solve scientific puzzles (which does not mean that participants in Borderland 3 do not contribute to citizen science or have a significant science learning experience). Unlike for SETI@home, the number of participants in Borderland 3 does not indicate a global trend towards greater public engagement with online citizen science. Thus, rather than a historical rise of citizen science over the last few decades, one is witnessing a transformation of public participation in science. The more traditional modes of public participation, such as amateur bird watching, are becoming increasingly visible online. Unlike some of the newer forms of participation, such as classifying galaxies for example, nature observation projects can count on pre-existing communities which have been well organized for more than a century [81,82]. This brings a further note of caution to any quantitative study that would infer a rise of public participation in science from online data alone. In some cases, at least, we may only be witnessing a digitalization of existing forms of public participation in science.

Public participation has been widespread in science since the Scientific Revolution, even though it is difficult to compare it to the current forms of participation, since science was not yet professionalized [18]. Yet, for most of science's history, there has been a tight connection between people predominantly engaged in nature inquiry and other people mostly involved in other activities [8]. In this respect, the twentieth century looks more like an exception, where the roles of professionals and amateurs have been particularly strongly delineated. Yet, even in the twentieth century, a few scientific data collecting projects have involved hundreds of thousands of participants, such as Operation Moonwatch operated by the Smithsonian Astrophysical Observatory between 1956 and 1975 to track satellites, which involved over 700,000 participants [83]. Similarly, the Baby Tooth Survey collected over 300,000 teeth donated by participants during the 1950s to study their level of Strontium 90 resulting from nuclear tests [84,85]. If public participation in science is effectively on the rise today, it is neither unprecedented (because public participation, under various names, never disappeared in the 20th century), nor a return to older forms of public participation (because science and society were in an entirely different configuration).

The third major question that our study allows us to investigate is the "democratization of science", another frequently stated goal of citizen science [86]. "Democratic" has several meanings, but most often, in this context, refers to making the scientific workforce less "elitist", i.e. more representative of the general population in terms of gender, age, ethnicity, and educational background. The data presented here shows that, in many cases, the online citizen scientist population is actually *less* representative of the general population with regards to gender and age. The gender imbalance in almost all online citizen science projects (over 90% in *computing*), for example, is greater than in professional science (60% male in the OECD in 2020). There is no simple explanation for the heavily male imbalance in online citizen science, as women have adopted other online technologies, such as Facebook, as much, if not more than men [87]. Women also comprised almost half of the gamer population in 2019 [88]. However, since the participants to SETI@home between 1999 and 2001 constitute the largest contribution (18%) to our dataset, the overall sex imbalance reflects, in part, the specificities of this project, which appealed particularly to men and took place at a time when computing was more strongly associated as a male activity [42]. As the population of participants to nature-based projects such as iNaturalist and eBird grows, which have close to gender parity, one expects the online citizen science population to become more balanced [60]. Numerous citizen science advocates are also making sustained efforts to make the participant population more inclusive and reach underrepresented populations [28,67], while other projects are more squarely focused on scientific outcomes.

In terms of age, participants in online citizen science projects are generally younger than the general population. However, citizen science projects include demographics that are not represented among professional scientists, obviously people under 18 or over 65-year-old. In that sense, the population is more representative of the general population.

Finally, in terms of education, the population of citizen scientists does not necessarily correspond to the popular depiction of the lay person discovering scientific research, since more than 80% of online citizen scientists, outside of *sensing* projects, have an occupation or a prior education in science or IT. The typical online citizen scientist, active in *computing* and *analyzing* projects, has a high level of scientific education, usually at least a bachelor's degree [26,28,32,33]. However, the fact that 20% of participants are *not* trained in science shows that this form of public participation constitutes a possible entry into research for lay people, offering a qualitatively different form of experience than outreach institutions such as museums. Also, embedding citizen science in popular online games, such as Borderlands 3, could contribute to broadening the social and educational background of participants (but not necessarily correct the gender imbalance).

Our quantitative study of public participation in science offers a critical outlook on the promises of online citizen science to democratize science and increase scientific literacy. It has also highlighted that behind aggregated numbers, there are large differences among projects, with some being highly successful in reaching underrepresented audiences or in producing scientific results that could not be achieved by professional science alone. Thus, if citizen science is not a "magic bullet" to cure science from some of its societal shortcomings, it remains one of many options to achieve some of these ideals. In citizen science, like in science more generally, greater inclusion could help align the goals and methods of research with the public interest. But that will only prove true in citizen science if the participants are given the necessary autonomy to determine these goals and methods [18].

Some of the metrics proposed here could prove valuable to gain a better understanding of online participation in science and beyond. For example, measuring the dynamics of active participants over time and per unit of time, the distribution of work within that population, its demographics, its spatial and temporal structure, are far more illuminating metrics than

counting the total number of accounts, especially since less than 1% of these are actually actively participating. We believe that providing more significant metrics of citizen science activity can contribute to fulfilling its future promises.

## Acknowledgments

We would like especially to thank Rick Bonney and Ken-ichi Ueda for providing critical comments on an initial version of the paper, as well as Grant Miller, Amy R. Sterling, Attila Szantner, Rom Walton, Tom Smith, Laurence Field, James Drews, Andy Browery, Brian Koepnick, Laurence Field, Juan A. Hindo, Michael Goetz, Bruce Allen, Matt Lebofsky, Massimo Giovannozzi, Oula Abu-Amsha, Helga Nowotny, Dominique Pestre, Christopher Wood, Kevin Reed, Gregory R. Bowman, and Sophie Czich for sharing anonymized data, contributing to data analysis, data visualization, and critical comments, as well as Thomas Mesaglio, for an exceptionally thorough and helpful review.

## Author Contributions

**Conceptualization:** Bruno J. Strasser, Elise Tancoigne, Jérôme Baudry, François Grey.

**Data curation:** Bruno J. Strasser, Elise Tancoigne, Jérôme Baudry, Steven Piguet, Helen Spiers, José Luis-Fernandez Marquez, David Anderson, Chris Lintott.

**Formal analysis:** Bruno J. Strasser, Elise Tancoigne, Jérôme Baudry, Steven Piguet, Helen Spiers, José Luis-Fernandez Marquez.

**Funding acquisition:** Bruno J. Strasser.

**Investigation:** Bruno J. Strasser, Elise Tancoigne, Jérôme Baudry.

**Methodology:** Bruno J. Strasser, Elise Tancoigne, Jérôme Baudry, Jérôme Kasparian.

**Project administration:** Bruno J. Strasser.

**Resources:** Bruno J. Strasser, Helen Spiers, José Luis-Fernandez Marquez, David Anderson, Chris Lintott.

**Software:** Elise Tancoigne, Steven Piguet, José Luis-Fernandez Marquez.

**Supervision:** Bruno J. Strasser.

**Validation:** Bruno J. Strasser.

**Visualization:** Bruno J. Strasser, Steven Piguet, Jérôme Kasparian, François Grey.

**Writing – original draft:** Bruno J. Strasser.

**Writing – review & editing:** Elise Tancoigne, Jérôme Baudry, Helen Spiers, Jérôme Kasparian, François Grey.

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
