## [Decision Letter · Decision Letter 0]

12 Jul 2023

PONE-D-22-35096Quantifying Online Citizen Science: Dynamics and Demographics of Public Participation in SciencePLOS ONE

Dear Dr. Strasser,

Thank you for submitting your manuscript to PLOS ONE. After careful consideration, we feel that it has merit but does not fully meet PLOS ONE’s publication criteria as it currently stands. Therefore, we invite you to submit a revised version of the manuscript that addresses the points raised during the review process.

We look forward to receiving your revised manuscript.

Kind regards,

Daniel de Paiva Silva, Ph.D.

Academic Editor

PLOS ONE

Journal Requirements:

We also thank the SNSF for generous funding through the ERC/SNSF Consolidator Grant (BSCGI0_157787), “The Rise of Citizen Science: Rethinking Public Participation in Science,” to B. J. Strasser at the University of Geneva. 

However, funding information should not appear in the Acknowledgments section or other areas of your manuscript. We will only publish funding information present in the Funding Statement section of the online submission form. 

This study was funded by the Swiss National Science Foundation (www.snf.ch) through the ERC/SNSF Consolidator Grant (BSCGI0_157787), “The Rise of Citizen Science: Rethinking Public Participation in Science,” to BJS at the University of Geneva. The funders had no role in study design, data collection and analysis, decision to publish, or preparation of the manuscript.

3. Please include your tables as part of your main manuscript and remove the individual files. Please note that supplementary tables (should remain/ be uploaded) as separate "supporting information" files

Additional Editor Comments:

Dear Dr. Strasser,

After this first review round, I had to send the manuscript for four different reviewers. The first two reviewers provided very dissonant decisions ("minor review" vs. "reject") so I had to search for two other evaluators of your study to reach a common ground on it. So, after four reviews, despite the initial "reject" decision, your manuscript received positive reviews from three reviewers (1 "accept" and 2 "minor review." Still, despite the rejection decided by Reviewer #2, I believe they have interesting points of view that should be integrated into the manuscript before publication. Therefore, I will grant you a major review decision on your manuscript to improve your manuscript accordingly.

After the thrre-month period I will grant you ends, I will ask you to please resubmit your manuscript with the included changes, along with a rebuttal letter informing the main changes that were made in the direction pointed out by all reviewers. In that rebuttal letter, you will also be able to explain and justify to the reviewers the reasons why this or that change was not included in the next version of your manuscript. As soon as you are finish, a new review round will comence, but, as already pointed out in the reviews, I believe your manuscript should be more than acceptable by the time you resubmit.

Sincerely,

Daniel Silva

Reviewers' comments:

Reviewer's Responses to Questions

**Comments to the Author**

1. Is the manuscript technically sound, and do the data support the conclusions?

Reviewer #1: Yes

Reviewer #2: No

Reviewer #3: Yes

Reviewer #4: Yes

2. Has the statistical analysis been performed appropriately and rigorously? 

Reviewer #1: Yes

Reviewer #2: N/A

Reviewer #3: I Don't Know

Reviewer #4: Yes

3. Have the authors made all data underlying the findings in their manuscript fully available?

Reviewer #1: Yes

Reviewer #2: Yes

Reviewer #3: Yes

Reviewer #4: Yes

4. Is the manuscript presented in an intelligible fashion and written in standard English?

Reviewer #1: Yes

Reviewer #2: No

Reviewer #3: Yes

Reviewer #4: Yes

5. Review Comments to the Author

Reviewer #1: The authors examined the global demographics of participation in citizen science projects were contributions can be provided online. They analysed the large amount of about 14 million participants worldwide (and not only in „Western“ countries) aiming to understand whether citizen science can contribute to the democratization of science. They concluded that due to the – in fact typical – overrepresentation of more science affine persons and the uneven distribution of participation (which is also typical for each social group ranging from sport associations to parents evenings at schools) online citizen science projects provide only a limited contribution to the democratization of science understood as the increasing participation of underrepresented groups.

Their observation of a peak on 2000 and the general timely dynamic is also interesting and worth thinking about a bit more. I may be linked to the general desire for more real social contacts, or a change in citizens scientists expectations but also technical opportunities.

The study is very interesting and scientificially sound. The database is disciplinarily and geographically mixed. The study provides hints how to monitor citizen science activities during a given time period – and therefore a tool to understand better the different interaction levels. I may also help to see social targets such as democratization or increasing scientific literacy a bit more differentiated.

Or to put it the other way round: State investments into education may also increase the value participants can gain from citizen science.

A learning point for citizen science project manager is that the „platformisation“ may help to keep participants with fluctuating interests

From my perspective the authors miss an important point in their discussion. Mainly data from online participation are used for machine learning which implies that the role of citizens and therefore their motivations may change, but also that the shown uneven distribution of contributers may impact the algorithms.

Reviewer #2: The authors present a study with important goals – to understand participation in “participatory” science, and they assembled an especially large database of participants from three types of projects: computing (providing computational power from their personal computers), analyzing (categorizing data or attempting to solve scientific problems), and sensing (reporting biodiversity observations). The authors presented several metrics to gauge participation across project types with a particular focus on gender, age, and country participation.

I found it very difficult to evaluate this research for two reasons, and so have declined to give it a full review. First, there was no place where they actually explained each of the projects (e.g., what is “BIOINC” – what is the difference between the different programs, e.g., SETI at home vs. climate, what is “FOLD-it” or the Multi-game universe”. I am familiar with some of these, but not all – and the authors should describe each). A table with every single project from which they extracted data would be extremely useful, including start and stop years. I also was not able to see results in all figures (especially Fig. 5) in that the graphic was so fuzzy, I couldn’t read the labels. Following the results was also made difficult because figures were at the end of the manuscript, while figure labels were dispersed throughout. I couldn’t keep hunting around for figure legends.

So, even though I am not providing a formal review, I do have some suggestions for the authors:

• Give more consideration to how you evaluate “computing” and other activities. Is offering computational resources to science being a citizen scientist? Or is it just donating resources? Are those resources even needed anymore? It seems that by 2017 (last year of data as per Fig. 3), participation was very low- and by 2021, only 0.1%. Does it matter that most of the people providing computational resources in the early 2000s were male and in rich countries (where it is more likely that a private citizen would have a computer?). Yet because these programs engaged so many more people, they appear to swamp out other types of participation. Participation that is, in my opinion, more relevant to “citizen” or participatory science.

• Your methods of designating gender and age seemed dubious at best. It seems safer to use this information only when you have some profile data. Determining gender and age based on user name (although it was not entirely clear that was what was being used – because the methods were not spelled out), seems risky.

Reviewer #3: The manuscript by Bruno Strasser et al. provides a novel and interesting insight into the worldwide development of online CS. Three important goals of CS are critically examined and discussed based on a large data set, thus making an important contribution to a realistic appraisal of online CS.

However, some of the methods, such as automatic coding, should be explained in more detail. It is also important to discuss the extent to which 'distributed computing' can be classified as citizen science. Some of the wording in the results section should also be reviewed, as it reads more like a discussion of the results.

-Abstract: Correct sentence: ‘These results highlight current challenges and the future potential of citizen science.’

-Methods:

-Because automatic coding is a central method in your paper, please introduce it in more detail in the beginning of the methods section, explaining its principles, aims and approaches. Specify the functioning – which package was used, is it AI/ algorithm based?

-l. 143-144. please discuss to which extent ‘computing’ i.e. providing computing capacity can be considered as a real CS activity- also taking into account the ten principles of CS (ECSA, 2015). What are participant motivations for computing? What benefits does computing have for participants?

-L. 184: please describe what the platform names.org was used for exactly

-L. 187ff: please reformulate this sentence to explain: how can you assign gender to a user profile that doesn’t exist?`

-L. 230: please explain why you used Lorenz curves, eg ‘to visualize xxx, we used…’

-Fig. 1: Add a legend to briefly explain three forms of participation. In methods part, you use the words: computing, analyzing, observing, in the figures, you write computing, sensing, analyzing- why this difference in terms?

-L. 287-342: several sentences read like you are already discussing the results in this section, instead of just reporting results in a neutral way, e.g.:

-Reformulate l.287-288: ‘This result, alone, already challenges the common narrative 287 about the recent “rise of citizen science” [37–39] or “rise of the citizen scientist” as an editorial in Nature put it’

-L 310 ‘misleading’

-L 329-330: ‘This demonstrates that the total number of accounts is a poor indicator of participant activity.’

-L. 341 ‘Hence, it is reasonable to say that there are under one million monthly

participants in citizen science year-round.’ -> this belongs into the discussion part. Moreover, please always make it clear that you are talking about online CS. The considerable offline CS activities worldwide are not included.

-L. 354 ff: you mention the Gini coefficient here, but this wasn’t mentioned or explained in the methods part

-L. 398: typo: cicizen science

-L. 415-419: this could fit into the introduction or discussion, rather than in results

-L.442: specify: ‘online citizen science population’ –you didn’t analyze offline CS activities

-L. 473: ‘as citizen science have valued the participation of ‘ordinary citizens’: specify - do you mean that CS actors or the CS community has valued xxx?

Discussion

-L.606ff: when discussing CS learning outcomes, it doesn’t make sense to talk only about participant numbers or visitor numbers. It is important to differentiate different forms of CS participation or engagement with science (Shirk et al. 2012)- eg computing projects (do they actually offer any learning outcome??) can hardly be compared to real CS activities where people receive training, interact with others and learn to interpret ecological data. An aspect to be considered is that a museum visit is a short-time event, while CS engagement can be a long-term, repeated learning opportunity. It would be helpful to include some of these thoughts into your discussion, also stating clearly that your data don’t comprise any information about learning outcomes

-L. 637: ‘Its value and significance can thus not be fully captured by the number of its practitioners alone.’ -> agree completely, could you highlight this sentence more in the abstract and beginning of discussion/ conclusion? It seems quite buried

Reviewer #4: An interesting paper that strives to characterise citizen science as a 'discipline' in its entirety rather than at an individual project level like most other studies. I think this paper will be heavily cited, and I admire the authors for tackling such a huge dataset. I also like that you sought to actually test whether 'entrenched'/much-parroted claims like 'citizen science is on the rise' are actually backed up by the stats rather than just accept it as true like all other studies (even if I don't fully agree with some of your rebuttals to these claims). My comments and suggestions are in the attached word file; most of them are minor corrections of typos or asking for clarification in places where I was confused or unclear about eg methodology, but a few important points here also:

1. I was unable to download or view any of your supplementary material. After clicking the download button, I waited for more than one hour and it did not work, all that happened was an infinite spinning circle over the download button, and every now and then the button would flash and say ‘request access’ instead of download. I also closed the page and tried again multiple times, including in multiple different browsers, but the download failed every time. I also emailed the link to multiple other people, and they couldn't download the material either. Could you double check whether you're able to download those files yourselves from the provided link, or whether the download is broken. It was unfortunate I couldn't access this material as I assume it would have answered some of my queries.

2. Perhaps my biggest concern is the extreme influence of a single outlier on your entire dataset, ie SETI@home. I completely acknowledge that you can't just omit it from your study because it's skewing results, and you do explicitly note its influence in your MS, but it seems like many of your major claims and findings are effectively being entirely driven by this single project. If you were to remove it, it seems like many of your reported trends/stats would be wildly different, and indeed you do already report this; for example, SETI@home had 90% males vs iNat 56% and eBird 51%, dramatically different figures. Growth peaked over 20 years ago (!!) for SETI@home, but eBird and iNat continue to grow exponentially. Perhaps this is too much additional work/would add too much complexity to your study, but I would be very interested in two parallel sets of results being presented, one with and one without SETI@home.

I think it's also a very relevant fact that as of March 2020, SETI@home has terminated sending new data to the public, and the citizen science aspect of the project has effectively gone into hibernation; you don't mention this at all in your paper, but it's an important point to acknowledge.

3. As an extension of my above comment, some of the major conclusions and findings that you report are effectively entirely based on the computing category (which of course includes SETI@home). Indeed, your findings for education/profession were based exclusively on computing projects, given you say "The analyzing projects on Zooniverse have no biographical profiles" and that "For sensing projects, such as iNaturalist or eBird, self-descriptions center on the affective dimension of amateur naturalist pursuit, and rarely mention occupation or education, preventing a comparison with the other projects". You do mention and cite earlier surveys on these two fronts, but of course this isn't the same as actually manually coding profiles from these projects like you did for the computing projects.

My worry is that your paper will be prone to 'mis-citations' by other authors that do not carefully read your paper and simply skim the abstract and results/discussion. Of course you have no control over that element and I don't expect you to take any responsibility for those cases, but I think you could perhaps pre-emptively address this issue by including more explicit explanations/caveats than what you currently have, or subtly adjusting some of your wording. For example, your abstract notes that "the vast majority of participants, male and female, have a background in science", but this is, in my opinion, misleading as a statement in isolation given, again, it is based entirely (from the perspective of your specific quantitative analysis) on 12,000 profiles (out of 14 million accounts, =0.09%!!) selected only from the computing category, and literally zero profiles from any of the analyzing or sensing projects.

6. PLOS authors have the option to publish the peer review history of their article (what does this mean?). If published, this will include your full peer review and any attached files.

Reviewer #1: No

Reviewer #2: No

Reviewer #3: No

Reviewer #4: **Yes: **Thomas Mesaglio

---

## [Author Response · Author response to Decision Letter 0]

8 Sep 2023

Response is included in the Cover Letter and the Response to Reviewers.

---

## [Decision Letter · Decision Letter 1]

10 Oct 2023

Quantifying Online Citizen Science: Dynamics and Demographics of Public Participation in Science

PONE-D-22-35096R1

Dear Dr. Strasser,

We’re pleased to inform you that your manuscript has been judged scientifically suitable for publication and will be formally accepted for publication once it meets all outstanding technical requirements.

Kind regards,

Daniel de Paiva Silva, Ph.D.

Academic Editor

PLOS ONE

Additional Editor Comments (optional):

Reviewers' comments:

Reviewer's Responses to Questions

**Comments to the Author**

1. If the authors have adequately addressed your comments raised in a previous round of review and you feel that this manuscript is now acceptable for publication, you may indicate that here to bypass the “Comments to the Author” section, enter your conflict of interest statement in the “Confidential to Editor” section, and submit your "Accept" recommendation.

Reviewer #3: All comments have been addressed

Reviewer #4: (No Response)

2. Is the manuscript technically sound, and do the data support the conclusions?

Reviewer #3: Yes

Reviewer #4: Yes

3. Has the statistical analysis been performed appropriately and rigorously? 

Reviewer #3: I Don't Know

Reviewer #4: Yes

4. Have the authors made all data underlying the findings in their manuscript fully available?

Reviewer #3: Yes

Reviewer #4: Yes

5. Is the manuscript presented in an intelligible fashion and written in standard English?

Reviewer #3: Yes

Reviewer #4: Yes

6. Review Comments to the Author

Reviewer #3: All open questions have been sufficiently addressed in the manuscript. The authors now state more clearly and also in the abstract what the limitations of the study are and provide explanations for the methods used. The role and potential of distributed computing is now also addressed in the introduction. This allows readers to better understand and interpret the results of the study.

Reviewer #4: Thank you for addressing all of my comments and suggestions in this revision, I think the paper is now well-framed and in good shape. I have attached another word document with comments/suggestions, but all of them regard minor typos or word changes; I have no major criticisms of the paper now, and am happy for the paper to be accepted after my minor suggestions are implemented.

Please note that for this review, all of my Line numbers refer to the tracked changes version of your manuscript, not the clean version. When I finished my review, I briefly skimmed the clean version, and noticed that there are at least a few small differences between the two submitted versions (clean and tracked changes). For example, in the clean version, Line 316 has the word "analyzing" as capitalised, but in the tracked changes version, this same word (at Line 321) is lower case. I am unsure which version is the most up to date, but I reviewed the tracked changes version. Thus, some of my comments/corrections may not apply, and you can disregard them if you have already made those amendments in the clean version (assuming it is more current than the tracked changes version).

Well done on the paper!

7. PLOS authors have the option to publish the peer review history of their article (what does this mean?). If published, this will include your full peer review and any attached files.

Reviewer #3: No

Reviewer #4: **Yes: **Thomas Mesaglio

---

## [Editor Report · Acceptance letter]

8 Nov 2023

PONE-D-22-35096R1 

Quantifying Online Citizen Science:
Dynamics and Demographics of Public Participation in Science 

Dear Dr. Strasser:

I'm pleased to inform you that your manuscript has been deemed suitable for publication in PLOS ONE. Congratulations! Your manuscript is now with our production department. 

Kind regards, 

on behalf of

Dr. Daniel de Paiva Silva 

Academic Editor

PLOS ONE